# Amyloid accelerator polyphosphate fits as the mystery density in α-synuclein fibrils

**Philipp Huettemann**[1‡¤], **Pavithra Mahadevan**[1‡], **Justine Lempart**[1], **Eric Tse**[2],
**Budheswar Dehury**[3], **Brian F. P. Edwards**[4], **Daniel R. Southworth**[2], **Bikash R. Sahoo**[5]*,
**Ursula Jakob**[1,6]*

1 Department of Molecular, Cellular and Developmental Biology, University of Michigan, Ann Arbor,
Michigan, United States of America, 2 Institute for Neurodegenerative Diseases, University of California San
Francisco, California, United States of America, 3 Department of Bioinformatics, Manipal School of Life
Sciences, Manipal Academy of Higher Education, Manipal, India, 4 Department of Biochemistry,
Microbiology, and Immunology, Wayne State University, Detroit, Michigan, United States of America,
5 Howard Hughes Medical Institute, University of Michigan, Ann Arbor, Michigan, United States of America,
6 Department of Biological Chemistry, University of Michigan Medical School, Ann Arbor, Michigan, United
States of America

¤ Current address: Department of Bioengineering and Therapeutic Sciences, University of California San
Francisco, California, United States of America
‡ These authors share first authorship on this work.
* bsahoo@umich.edu (BRS); ujakob@umich.edu (UJ)

pbio.3002650

Institute of Technology, UNITED STATES OF
AMERICA

**Data Availability Statement:** All data files are
available from the Mendeley database (https://data.
mendeley.com/datasets/3ybktpvbsd/2).

## Abstract

Aberrant aggregation of α-Synuclein is the pathological hallmark of a set of neurodegenerative diseases termed synucleinopathies. Recent advances in cryo-electron microscopy have led to the structural determination of the first synucleinopathy-derived α-Synuclein fibrils, which contain a non-proteinaceous, "mystery density" at the core of the protofilaments, hypothesized to be highly negatively charged. Guided by previous studies that demonstrated that polyphosphate (polyP), a universally conserved polyanion, significantly accelerates α-Synuclein fibril formation, we conducted blind docking and molecular dynamics simulation experiments to model the polyP binding site in α-Synuclein fibrils. Here, we demonstrate that our models uniformly place polyP into the lysine-rich pocket, which coordinates the mystery density in patient-derived fibrils. Subsequent in vitro studies and experiments in cells revealed that substitution of the 2 critical lysine residues K43 and K45 with alanine residues leads to a loss of all previously reported effects of polyP binding on α-Synuclein, including stimulation of fibril formation, change in filament conformation and stability as well as alleviation of cytotoxicity. In summary, our study demonstrates that polyP fits the unknown electron density present in in vivo α-Synuclein fibrils and suggests that polyP exerts its functions by neutralizing charge repulsion between neighboring lysine residues.

## Introduction

The 140 amino acid protein α-Synuclein (α-Syn), which is widely expressed in neuronal and non-neuronal cells in the brain, acts as the main pathological agent in a number of diseases,

**Funding:** This work was supported by the National Institute of Health grant R35 GM122506 to U.J. Salary was paid from NIH for P.H., P.M and J.L. The salary for B.S. was provided by the Howard Hughes Medical Institute. The funders had no role in study design, data collection and analysis, decision to publish, or preparation of the manuscript.

**Competing interests:** UJ is a member of the PLOS Biology Editorial Board.

**Abbreviations:** LBD, Lewy body dementia; MD, molecular dynamics; MGL, Molecular Graphics Laboratory; MSA, multiple system atrophy; PD, Parkinson's disease; PME, particle mesh Ewald; SDS, sodium dodecyl sulphate; TEM, transmission electron microscopy; ThT, Thioflavin T; WT, wild type.

collectively termed synucleinopathies [1]. In healthy human brains, α-Syn is a monomeric, highly soluble protein that promotes membrane curvature [2], and plays a role in synaptic trafficking and vesicle budding [3,4]. Although considered predominantly intrinsically disordered, more recent work demonstrated that the N-terminal region of α-Syn acquires α-helical conformations and interacts with membranes [2,5]. In synucleinopathies, such as Parkinson's disease (PD), multiple system atrophy (MSA), or Lewy body dementia (LBD), however, α-Syn deposits as insoluble β-sheet rich fibrils. The formation of these fibrils is initiated by α-Syn monomers that adopt a β-sheet structure (i.e., amyloid fold). These monomers subsequently self-associate to form oligomeric nuclei and ultimately deposit as larger oligomers, protofibrils, and fibrils in the cytosol of neuronal cells. The mature fibrils consist of one or more helically twisted protofilaments as each α-Syn monomer is slightly offset from the neighboring unit. In some synucleinopathies, such as MSA, 2 protofilaments, each consisting of α-Syn monomers adopting the prototypical MSA-fold (MSA-polymorph; pdb: 6XYO) associate into double-twisted fibrils [6] (S1A Fig). In others, such as LBD, α-Syn monomers adopt the Lewy-fold and deposit as a single protofilament with a reduced helical twist (LBD-polymorph; pdb: 8A9L) (S1B Fig) [7]. Neurotoxicity is thought to arise from oligomeric species disrupting membranes [8,9], mitochondrial function [10,11], and/or protein clearance pathways [12]. The release of toxic α-Syn species from infected cells and their uptake into healthy neurons appears to contribute to the spreading of the disease within different regions of the brain [13–15].

Much effort has been spent to identify molecules that alter the aggregation kinetics of amyloid fibril formation [16]. One such modifier is polyphosphate (polyP), a polyanion composed of up to 1,000 phospho-anhydride linked phosphates. PolyP is ubiquitously present in every cell and tissue studied, including the brain, where polyP levels are in the micromolar amounts (in Pi-units) and chain lengths are, on average, about 800 residues [17]. It remains unclear how these long chain lengths are synthetized and regulated in mammalian cells [17]. Previous studies demonstrated that polyP is highly effective in stimulating disease-associated fibril formation in vitro [18]. PolyP significantly accelerates α-Syn fibril formation at physiologically relevant protein and polyP concentrations and protects neuronal cells against amyloid toxicity by interfering with the membrane association and uptake of α-Syn oligomers [18,19]. Mechanistic studies showed that polyP nucleates smaller α-Syn oligomers, alters fibril morphology, and increases fibril stability [20]. Transmission electron microscopy (TEM) [19] and cryo-EM analysis revealed that recombinant α-Syn fibrils formed in the presence of polyP consist of a single protofilament with a reduced helical twist, which has hampered all attempts in obtaining a high-resolution cryo-EM structure, leaving the question open as to how polyP interacts with α-Syn fibrils.

Intriguingly, atomic resolution structures of patient-derived but not in vitro-derived α-Syn fibrils [21] (S1C Fig) uncovered the presence of a "mystery density," which is embedded within a central core formed by the 2 protofilaments of the MSA-polymorph [6] (S1A Fig) or in a surface exposed groove in the Lewy-fold single protofilament [7] (S1B Fig). In each case, the density is associated with a core group of positively charged amino acids, i.e., K43, K45, and H50, and includes 2 additional lysine residues K32 and K34 in the single protofilament structure. Described as non-proteinaceous and highly negatively charged, this mystery density compensates for 3 to 4 positive charges per rung [6]. A similar density surrounded by a pocket of lysine residues was recently identified in TDP43 [22] as well as in patient-derived Tau polymorphs [23–25], another amyloid, whose fibril formation is significantly accelerated by polyP [18,26]. Given the unique size and charge requirement of the unknown electron density, we now hypothesized that polyP might be occupying this binding pocket in α-Syn fibrils. Blind docking and molecular dynamics (MD) simulations combined with mutagenesis and polyP binding studies confirmed that polyP interacts with α-Syn fibrils via the lysine-rich core that interfaces

with the observed "mystery density." Substitution of these lysine residues abolished the effects of polyP on α-Syn fibril formation, fibril morphology, and cytotoxicity. These results suggest that polyP binds to this lysine-rich core and accelerates filament formation by minimizing charge-repulsions between neighboring lysine residues in α-Syn fibrils. We propose that polyP fits the mystery density observed in patient-derived fibrils.

## Results

### Identification of polyP binding sites by molecular docking and MD simulations

Characterization of the molecular mechanism by which physiologically relevant modifiers such as polyP bind to α-Syn, modulate their aggregation kinetics, and affect fibril morphology is critical for discovering potential new treatments against synucleinopathies. Our previous results revealed that polyP binds to oligomeric species and pre-formed fibrils but shows no significant affinity for α-Syn monomers [19]. To gain a better understanding as to how polyP interacts with α-Syn, and to assess whether the previously observed density in patient-derived fibrils might correlate with polyP binding, we used computational methods, including molecular docking and atomic-level MD simulations. These approaches have been successfully used to characterize the interaction of α-Syn with small molecules, membranes, metal ions, and other proteins [27–29]. To probe the interaction between polyP and α-Syn, we used energy-minimized modeled structures of a 5-mer (polyP-5) or 14-mer polyP (polyP-14) (see Methods) and tested in silico binding to 3 different fibrils, whose atomic resolution structures have been solved (Table 1); the MSA patient-derived polymorph 6XYO [6] (S1A Fig); the LBD patient-derived polymorph 8A9L [7] (S1B Fig); and the in vitro-derived polymorph 6H6B [21] (S1C Fig), which does not contain any additional density according to the cryo-EM structure.

In addition, we investigated binding of polyP-14 to a single β-sheet rich monomer of the Lewy-fold polymorph 8A9L as well as a single α-helical α-Syn monomer 1XQ8 [30] (S2A Fig). For the polymorphs consisting of 2 protofilaments (i.e., 6XYO and 6H6B), we used a double filament structure consisting of 10 α-Syn monomers (5 per protofilament) while for the single filament structure 8A9L, we used 3 α-Syn monomers (S2A Fig). We considered a polyP length of either 5 or 14 to be best suited for studying molecular interactions with these small α-Syn fibril structures.

AutoDock Vina blind docking simulation results showed that polyP-14 binds to residues S42, T44, E46, G47, V66, and G68 of the β-sheet rich monomer 8A9L with an affinity of −4.8 kcal/mol and to charged N-terminal residues (K6, K10, K12, E13) of the α-helical α-Syn monomer 1XQ8 with an even weaker affinity of −3.3 kcal/mol (S2B Fig). In contrast, the simulations yielded a stronger binding affinity for the protofilament structures; i.e., −5.8 kcal/mol for the binding of polyP-14 to the trimeric single protofilament structure of the Lewy-fold polymorph 8A9L; −6.0 kcal/mol for the binding to the double-twisted protofilament of

**Table 1. Fibril structures used in our docking studies and molecular dynamics simulations.**

| α-Syn structure | PDB ID | Origin | "Mystery Density" | Structural features | Reference |
|---|---|---|---|---|---|
| Fibril | 6H6B | In vitro | no | Two protofilaments | [21] |
| Fibril | 6XYO | MSA patient | yes | Two protofilaments | [6] |
| Fibril | 8A9L | LBD patient | yes | One protofilament | [7] |
| Monomer | 8A9L | LBD patient | yes | β-sheet rich | [7] |
| Monomer | 1XQ8 | In vitro | no | Micelle bound α-helical | [30] |

polymorph 6XYO and −7.4 kcal/mol for the double-twisted protofilaments of in vitro generated polymorph 6H6B (S2B Fig). Molecular interaction network analysis revealed that the best docked position of anionic polyP-14 coincides with the strong electron density in 6XYO, which is anchored by the side chains of the 3 charged residues K43, K45, and H50 from both protofilaments (Figs 1A, S1A, and S3A). In the docking model, the side chains of K43 and K45 form a symmetrical hydrogen bonding network that allows polyP-14 to adopt a U-shaped conformation, folding back into the central core and effectively filling the central cavity with 2 adjacent polyP chains (Fig 1B). Further analysis of the bonding network revealed that each Pi-unit is in contact with 3–4 α-Syn molecules from both protofilaments while the free phosphate units at the ends of the polyP-14 structure interact with the first α-Syn molecules (chains G and H) in each protofilament (Fig 1B, pink lines). In addition to K43 and K45, we found that residues H50 and Y39 form electrostatic pi-anion and hydrogen bond interactions with polyP-14, respectively (Fig 1A). Upon positioning the modeled polyP-14 structure into the density map of the cryo-EM structure of polymorph 6XYO, we noted that 7 continuous phosphate units map the "mystery density" in the cryo-EM structure (Fig 1C). The remaining 7 phosphate units of polyP-14 fold back according to our docking studies, occupying an empty space in the core of the fibril. We assume that in longer α-Syn fibrils, polyP remains fully extended to form the maximal number of favorable interactions with lysine residues. In the case of polymorph 6H6B, our docking simulations also placed polyP-14 into close association with K43 and K45 albeit with slightly smaller binding energy (−5.6 kcal/mol) as compared to an adjacent L-shaped groove region (−7.4 kcal/mol) (S2B Fig). In the case of the trimeric structure of the single protofilament polymorph 8A9L, however, AutoDock generated the best docked position for polyP-14 in a region adjacent to the mystery density (S2B Fig). Given our previous observation in 6XYO that polyP-14 binds to residues of multiple α-Syn subunits of both protofilaments, we now wondered whether this offsite binding of polyP-14 in the single-face 8A9L trimer might be due to a lack of sufficient binding interface that could accommodate longer polyP molecules. To test this idea, we used the shorter polyP-5 molecule for further docking analysis. Indeed, out of 9 best-dock positions, we obtained a docking position for polyP-5 that falls exactly within the mystery density, allowing polyP-5 to make contacts with K32, K34, K43, and K45 and a binding energy of −4.6 kcal/mol (S3B Fig). These results show that polyP is also a plausible fit for the density observed in the single protofilament structures of patient-derived fibrils.

To apply an unbiased approach and track the ligand binding site in a physiological environment [31], we conducted molecular interaction studies in an aqueous solution using atomistic MD simulations. Under these conditions, we found no interactions between polyP-5 and α-Syn monomer 8A9L (Fig 1D). Moreover, independent of the absence or presence of polyP, the monomer transitioned into a more disordered-like conformation during the 100 ns time period of the simulation. These results were consistent with our experimental data, which failed to show any significant interaction between polyP and purified α-Syn monomers [19]. We then tested the interaction between polyP-14 and the α-Syn polymorph 6XYO in an aqueous environment under near physical conditions using 2 independent MD programs and force fields, i.e., Gromacs [32] (Charmm36) [33] and Desmond (OPLS4) [34,35]. In support of our docking experiments, we found that polyP-14, which was placed >1 nm apart from the double-twisted MSA-polymorph 6XYO in Gromacs, interacts with residues K43 and K45 (both via hydrogen and electrostatic interactions) (Fig 1E). MD snapshots derived at 50 ns showed a U-shaped polyP that forms 5 salt-bridges, 3 hydrogen bonds, and 11 electrostatic interactions with residues K43 and K45 present in the D/F α-Syn subunits of one protofilament and the C subunit from the other protofilament (Fig 1F). H50 forms 2 hydrogen bonds, one with the oxygen center of phosphate 7 and the other one interacting with the terminal

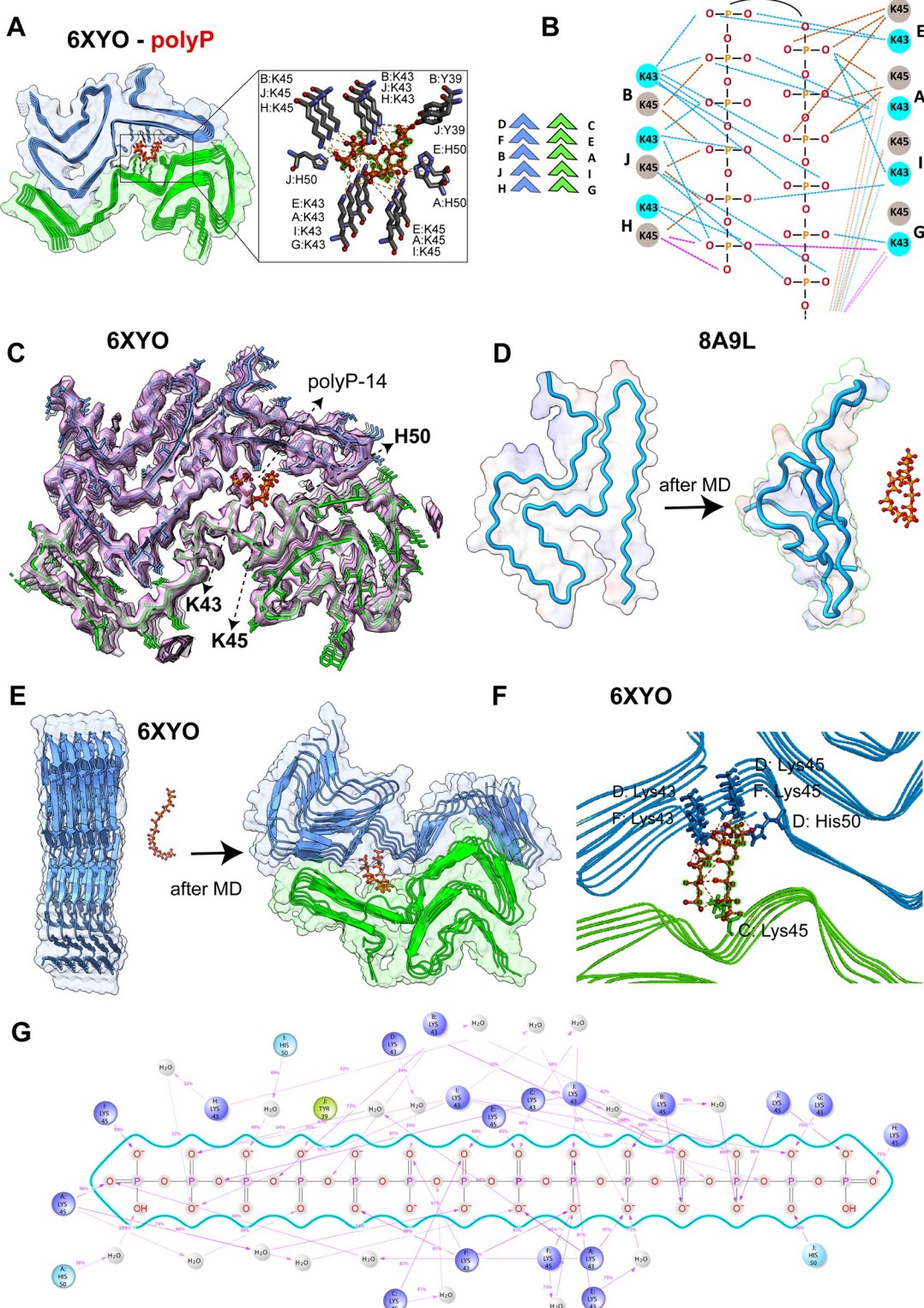

**Fig 1. Atomistic interactions between polyP and α-Syn fibrils. (A)** Molecular docking simulations showing polyP-14 binding sites in α-Syn fibrils (PDB ID: 6XYO). α-Syn and polyP-14 molecules are shown as surface and ball-and-stick, respectively in the cartoon structure. (**B**) A schematic model showing the bonding network (dashed lines) for lysine residues in α-Syn molecules distributed on the upper (D-F-B-J-H, blue) and lower (C-E-A-I-G, green) protofilaments of the 6XYO fibril structure. Shown are interactions (represented by pink lines) representing 1 free phosphate unit interacting with the first 2 α-Syn molecules facing

opposite in each protofilament (chains G and H). (C) Cartoon structure of polyP-14 (red) docked against the 6XYO electron density (EMD-10650) showing the mystery high-charge density core is occupied by 7 phosphate units in polyP-14. (D, E) MD snapshots showing the polyP interaction with α-Syn monomeric single filament structure 8A9L (D) and fibril protofilament 6XYO (E) before and after 50 ns MD simulation in Gromacs. (F) Cartoon structure of the final MD snapshots obtained from 50 ns all-atom MD simulation in Gromacs showing the α-Syn residues in 6XYO interacting with polyP-14 (in stick). Chain names for individual residues are shown before each colon. (G) Interaction pattern of polyP-14 with α-Syn residues (6XYO fibril structure) obtained from 100 ns MD simulation in Desmond. The underlying data can be found in Mendeley (see data statement for details).

oxygen, thus stabilizing the U-shaped polyP structure (Fig 1F). These MD results nicely recapitulated the in-vacuum docking simulation results. Interaction network analysis obtained from 100 ns MD simulation of the α-Syn fibril structure 6XYO and polyP-14 complex in Desmond revealed that interactions between the phosphate units and α-Syn K43 and K45 residues in chains A-I and H-J after 100 ns MD simulations occurred >75% to 99% of the simulation time. Complex analysis reproduced several short-lived long-range hydrogen bonds for free oxygen atoms and oxygens that link phosphate atoms to form phosphor-anhydride bonds (Fig 1G). We found that hydrogen bonds defined within a distance of 2.5 Å and involving K43 and K45 residues predominate over ionic interactions (S4A Fig). Notably, we observed a strong hydrogen bond network mediated by water molecules (water-bridges), suggesting that a hydrated bio-interface for polyP within the strong density core may play a crucial role in binding to these insoluble fibrils. Interaction analysis shows polyP-14 forms an average of approximately 48 specific intermolecular contacts with α-Syn fiber structure 6XYO primarily mediated by hydrogen bonding and hydrophobic interactions (S4B Fig). Contact timeline analysis further revealed that residues K43 and K45 in chains A, B, F, H, I, and J have a higher contact timeline with polyP than the other lysine residues, suggesting that these residues are crucial for polyP binding. MD simulations using the in vitro generated polymorph (6H6B) yielded similar results and, as before, depicted polyP to interact with K43 residues in 2 α-Syn chains distributed in one of the protofilaments with additional interactions that involve residues L38, Y39, and S42 (S5 Fig). Though the 50 ns MD simulation using Gromacs did not completely recapitulate the binding interface of polyP as observed in our AutoDock simulation (S5 Fig), both the programs indicate the possibility of polyP to bind regions surrounded by residues K43 and K45.

To computationally assess if polyP binding causes any structural or dynamic changes in α-Syn-fibrils, we calculated the root mean square deviation of Cα-atoms in the α-Syn polymorph 6XYO in complex with polyP-14. The deviation was <0.4 Å, suggesting that a stable plateau has been achieved for the complex during the 100 ns MD simulation (S4C Fig). PolyP-14 also showed a hydrated and compact structure within the strong density core in the α-Syn fibril (S4D Fig). Intriguingly, 3 α-Syn molecules in 6XYO (Chains G, C, E) and 2 in 6H6B (Chains C and G) showed stronger root means square fluctuation in the central region (residues ~ 50 to 60) in the presence of polyP as compared to α-Syn fibers alone (S6 Fig). Based on these results, we now speculate that this increase in root means square fluctuation may indicate an onset of disorder that could destabilize and alter the molecular packing or both within the β-stacked fibril morphology and might explain why polyP binding to in vitro formed double-twisted fibrils leads to their dissociation into individual protofilaments [19]. In summary, these in silico results strongly supported our hypothesis that polyP constitutes the mystery density observed in patient-derived fibrils.

## K43A, K45A mutations accelerate α-Syn fibril formation and eliminate effects of polyP

To probe if the core of positively charged residues is indeed involved in the binding to polyP, we replaced K43, K45, and H50 individually with alanine. In addition, we constructed 1

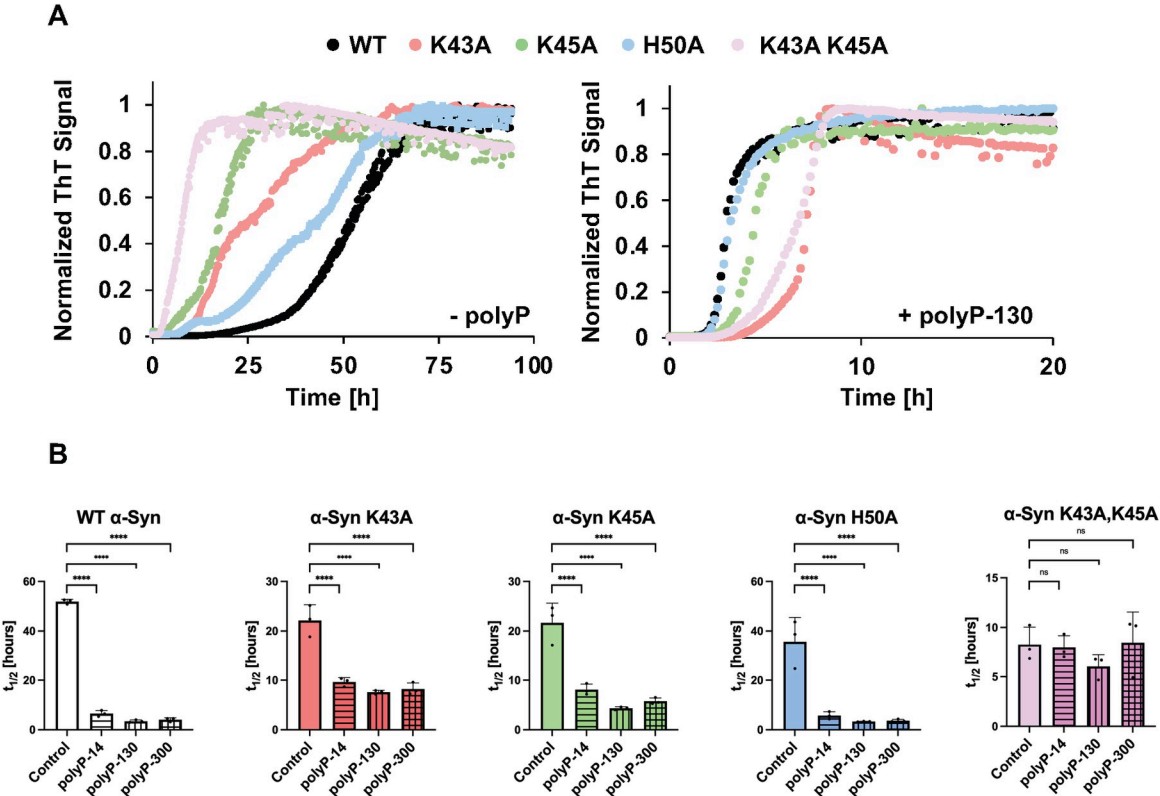

**Fig 2. Effect of polyP on the kinetics of WT and mutant α-Syn fibril formation.** (**A**) ThT fluorescence curves of 100 μM α-Syn wild-type and mutant proteins in the absence (left panel) or presence of 500 μM polyP-130 (in Pi-units) in 40 mM KPi, 50 mM KCl (pH 7.5). $n = 3$ and a representative graph is shown. (**B**) Respective half-lives ($t_{1/2}$) of fibril formation. The mean of 3 experiments ±SD is shown. Statistical analysis was prepared with one-way ANOVA (ns $P$-value >0.05, **** $P$-value <0.0001) comparing each polyP sample to the control sample (absence of polyP). The underlying data can be found in Mendeley (see data statement for details).

mutant variant that lacked the 2 lysine residues (α-Syn$^{K43A,K45A}$) as those appeared to play the most central role in polyP binding according to our in silico results. We purified the mutant variants and tested their in vitro fibril formation properties using Thioflavin T (ThT) fluorescence in the absence and presence of a 5:1 ratio of polyP to α-Syn. This ratio was found to be optimal in accelerating α-Syn fibrillation in prior in vitro studies [18], and consistent with reports that the mammalian brain contains between 50 and 100 μM in Pi units [36] and between 5 and up to 50 μM α-Syn [18,37]. As previously shown, even at very high concentrations (i.e., 100 μM), physiological temperatures (37°C), and the presence of nucleators (i.e., glass beads), α-Syn fibril formation proceeds very slowly with a halftime ($t_{1/2}$) of approximately 50 h. In contrast, in the presence of a physiological chain length polyP (i.e., polyP-130) and a ratio of polyP (in Pi-units) to α-Syn monomer of 5:1, fibril formation is significantly accelerated and proceeds with a $t_{1/2}$ of less than 3 h (Fig 2A, compare black traces in right and left panels). All 4 mutant proteins followed a wild-type–like sigmoidal ThT fluorescence curve, which maps the conversion of monomeric α-Syn into higher molecular weight oligomers and ultimately into mature fibrils. While the mutation of H50 showed little effect on the $t_{1/2}$ (approximately 40 h), substitution of either K43 or K45 with alanine reduced the $t_{1/2}$ by about 2-fold ($t_{1/2}$ approximately 20 h). When present in combination (i.e., α-Syn$^{K43A,K45A}$), we observed an even more drastic acceleration in fibril formation as reflected in a $t_{1/2}$ of 8.3 h. These results provided first evidence that this cluster of positively charged amino acids in wild-type α-Syn contributes to the slow rate of in vitro fibril formation.

Next, we conducted the same experiments in the presence of 500 μM polyP-130. While the α-Syn H50A mutant showed again wild-type–like behavior and a polyP-130-mediated rate increase of approximately 10.8-fold, both α-Syn K43A and K45A single mutants were less sensitive to the presence of polyP-130 and only increased their rate of fibril formation by about 2.5-fold. The α-Syn$^{K43A,K45A}$ double mutant, in contrast, showed no significant rate increase with polyP-130 and its $t_{1/2}$ of fibril formation remained at approximately 8 h. We obtained similar results when we used different chain lengths of polyP (Fig 2B). In each case, presence of polyP significantly accelerated the fibril formation of wild-type and α-Syn H50A but failed to stimulate fibril formation of the α-Syn$^{K43A,K45A}$ mutant. These results concurred with our in silico results, and strongly suggested that the 2 lysine residues, K43 and K45, which coordinate the "mystery density" in patient fibrils, are essential for the polyP-mediated stimulation of fibril formation. Our findings furthermore suggested that polyP binding accelerates fibril formation by neutralizing the positive charges of α-Syn's core lysine residues.

## Substitution of K43 and K45 abolishes polyP binding in vitro

To directly test whether the substitution of Lys to Ala at positions 43 and 45 is sufficient to prevent binding of polyP in vitro, we incubated wild-type (WT) α-Syn and α-Syn$^{K43A,K45A}$ (each at 100 μM) for 48 h in the presence of increasing concentrations of polyP-130. We then pelleted the fibrils and washed them with high salt buffer to remove any non-specifically bound polyP. To determine how much polyP remained bound to the fibrils after this procedure, we resuspended the protein pellets in 1 M hydrochloric acid, which not only dissociates α-Syn fibrils but also hydrolyzes any remaining polyP into inorganic phosphate that can be spectroscopically measured using the molybdate assay [38]. As we increased the polyP levels during fibril formation, we found increasing amounts of polyP-130 pelleted with WT α-Syn fibrils but not with α-Syn$^{K43A,K45A}$ mutant fibrils, unless very high concentrations of polyP were used during the incubation (Fig 3A). For polyP-130 concentrations at or below 250 μM, the Pi concentration recovered from the WT α-Syn protein pellet corresponded directly to the polyP concentration used in the incubation reaction. Any further increase in the polyP concentration, however, did not result in any higher levels of polyP bound to the fibrils, suggesting that binding saturation is reached at a ratio of 2 phosphate units to 1 WT α-Syn monomer. Our finding that substitution of the 2 lysine residues prevents robust binding of polyP clearly demonstrated that these 2 residues are indeed crucial for the interaction of WT α-Syn and polyP.

## Effects of K43A, K45A mutations on α-Synuclein fibril morphology and stability

Earlier studies revealed that the addition of polyP, either during the process of α-Syn fibril formation or to fully formed mature fibrils, significantly changes their morphology [18,19]. While in vitro formed WT α-Syn fibrils are highly heterogeneous and consist of 2 and more double-twisted protofilaments (average diameter of approximately 22 nm) with varying degrees of helical twists, poly-P associated α-Syn fibrils typically consist of a single, straight filament (average diameter of approximately 14 nm) with little detectable twist (Fig 3B). To investigate how substitution of the 2 critical lysine residues K43 and K45 affects fibril morphology, we negatively stained WT α-Syn and α-Syn$^{K43A,K45A}$ fibrils and analyzed them by TEM. On average, we found that α-Syn$^{K43A,K45A}$ fibrils have a similar morphology to WT α-Syn fibrils albeit with a slightly smaller average diameter of approximately 19 nm (Fig 3B). Moreover, and as expected based on our previous results, the presence of polyP during fibril formation had no significant effect on the fibril morphology of the mutant variant.

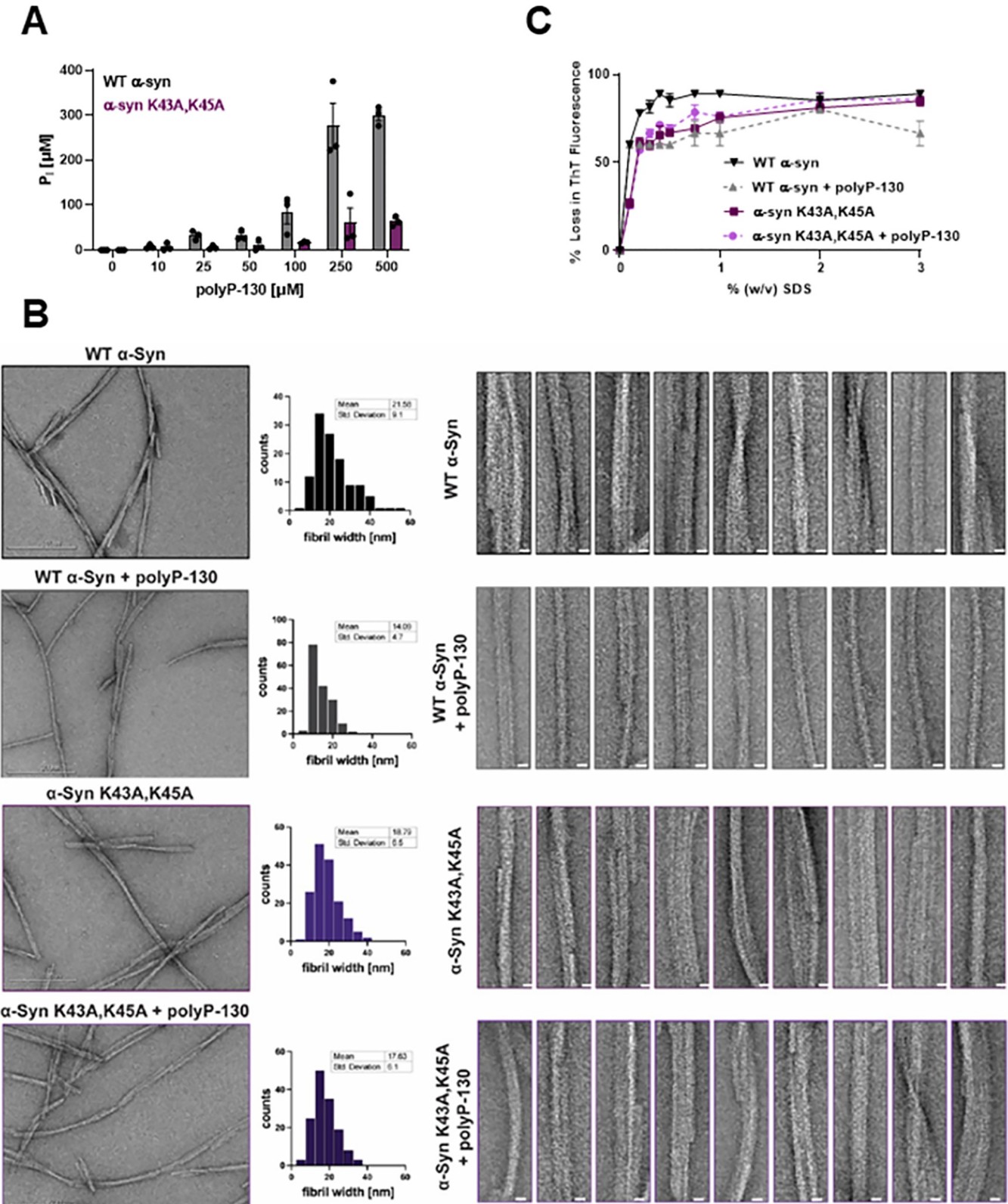

**Fig 3. Effects of K43, K45 substitution on polyP binding, fibril stability and morphology.** (**A**) Fibrils of 100 μM WT α-Syn or α-Syn[K43A,45A] were formed in the absence or presence of increasing amounts of polyP-130 for 48 h. Fibrils were pelleted, washed with high salt buffer and digested in 1 M hydrochloric acid

to hydrolyze all bound polyP. Free Pi was measured using molybdate. (**B**) TEM of 100 μM WT α-Syn or α-Syn$^{K43A,K45A}$ fibrils formed in the absence or presence of 500 μM polyP-130. Frequency distribution plots of the fibril width distributions and the mean fibril width and standard deviation are given in the middle panel. Ten representative filaments are presented on the right for each condition (scale bar is 10 nm). At least 100 fibrils per condition were measured across 3 separate experiments. (**C**) Fibrils were prepared in the absence or presence of 500 μM polyP-130 as before. Fibrils of WT α-Syn (black/gray) or α-Syn$^{K43A,K45A}$ mutant (light and dark purple) fibrils were supplemented with ThT and incubated with increasing concentrations of SDS at 37°C for 5 min. Changes in ThT fluorescence were recorded; $n = 3$; mean ± SD is shown. The underlying data can be found in Mendeley (see data statement for details).

In addition to morphological changes, WT α-Syn fibrils formed in the presence of polyP are less prone to shed off smaller oligomers [18], consistent with polyP binding stabilizing the fibrils. To test the extent to which polyP binding, polyP-mediated charge screening or both, contribute to this stabilizing effect, we prepared WT α-Syn and α-Syn$^{K43A,K45A}$ fibrils in the absence or presence of polyP-130 as before. We pelleted the fibrils, washed them with high salt buffer to remove any non-specifically bound polyP, and incubated the fibrils with ThT in the presence of increasing concentrations of SDS [39]. After 5 min of incubation, we monitored the changes in ThT fluorescence as a measure of fibril destabilization and dissociation. As shown in Fig 3C, we found that WT α-Syn fibrils are highly SDS sensitive. The ThT fluorescence readily decreased to less than 10% upon incubation of the fibrils in 0.5% w/v SDS. The presence of polyP significantly increased the stability of fibrils towards SDS-mediated dissociation, as indicated by the substantially higher concentrations of SDS needed to decrease the ThT signal to the same extent (Fig 3C). The lack of both K43 and K45 also increased the stability of the mutant protein, suggesting that not the binding of polyP chains per se but their ability to charge-screen the cluster of positive charges might contributes to the stabilization of α-Syn fibrils. Consistent with this model, we found that polyP showed no additional effect on the stability of α-Syn$^{K43A,K45A}$ fibrils. We obtained similar results when we monitored the SDS-mediated destabilization of the fibrils by centrifugation (S7 Fig).

## PolyP binding suppresses α-Syn cytotoxicity

Our studies indicated that the 2 lysine residues, K43 and K45, if not appropriately charge-neutralized, unfavorably affect α-Syn fibril formation and stability. To investigate how these features contribute to α-Syn toxicity, we tested the effects of sonicated WT α-Syn and α-Syn$^{K43A,K45A}$ fibrils on neuronal cell toxicity. We prepared fibrils in the presence or absence of polyP-130 as before, and, upon sonication, added them to freshly differentiated SH-SY5Y neuroblastoma cells. After 24 h of incubation, we measured total cell counts via Hoechst staining and dead cells using propidium iodide staining (Fig 4A). As shown before [18,19], incubation of SH-SY5Y cells with sonicated WT α-Syn fibrils caused significant cytotoxicity, which was largely eliminated in the presence of polyP (Fig 4B, compare dark gray columns). The toxicity of α-Syn$^{K43A,K45A}$ fibrils was similar to WT Syn fibrils in the absence of polyP-130 yet remained unaffected by the presence of polyP (Fig 4B, compare purple columns). These results reinforced our model that polyP binding and concomitant morphological alterations contribute to the observed reduction in WT α-Syn fibrils fibril toxicity.

## Discussion

The search for modifiers suitable to affect in vivo amyloid fibril formation and reduce toxicity remains a primary task in the fight against amyloid-associated pathologies. Previous work from our lab and others revealed that polyP, a naturally occurring polyanion, is highly effective in stimulating α-Syn, Aβ, and Tau amyloid fibril formation in vitro, altering α-Syn fibril morphology and reducing α-Syn and Aβ-amyloid-mediated toxicity in neuronal cell culture models [18,19]. These effects become more pronounced as the polyP chain lengths increase, a

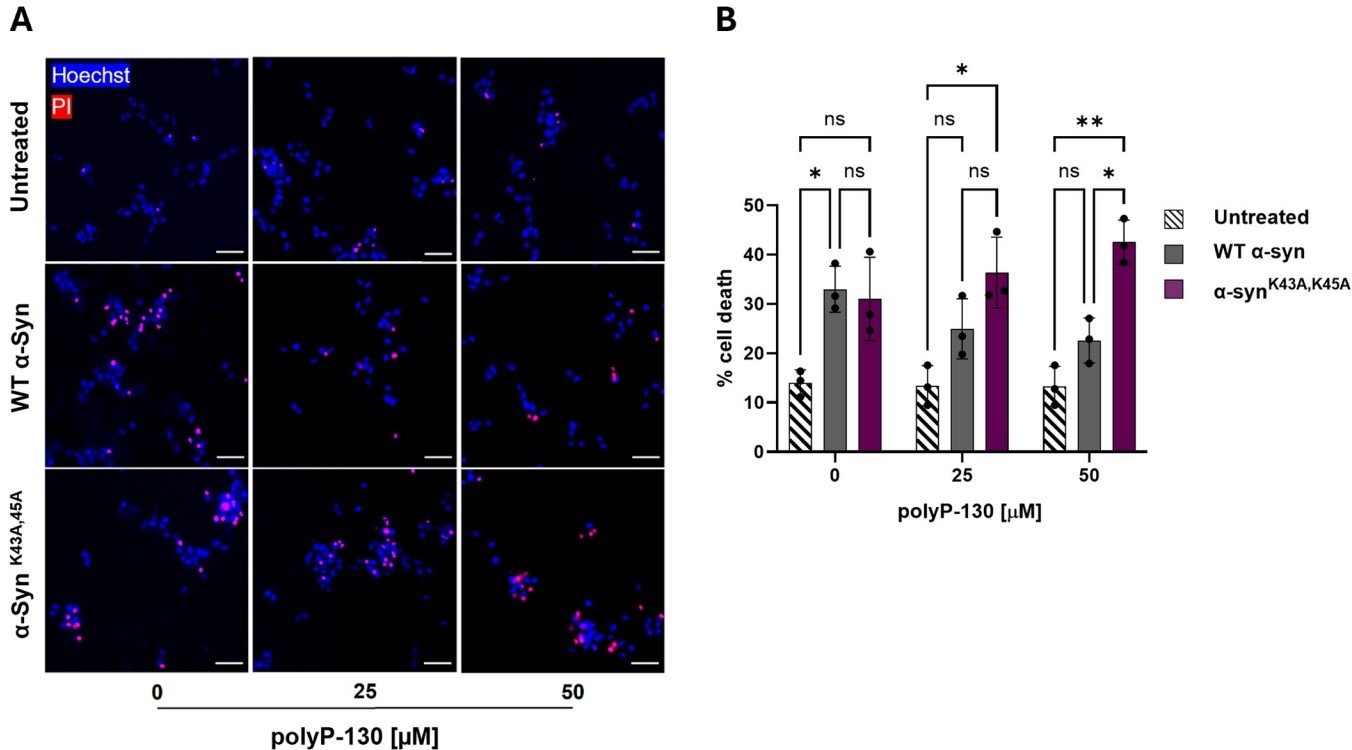

**Fig 4. Effects of polyP binding on cytoxicity of α-Syn. (A)** Representative overlays of differentiated SH-SY5Y cells after 24 h treatment with 4 μM sonicated fibrils of WT α-Syn or α-Syn[K43A,K45A] mutant in absence or presence of of polyP-130. Cells were stained with Hoechst to indicate all cells present in the image (blue), and with propidium iodide (PI) to identify the dead cells (red). Cells positive for both stains are observed in magenta and considered dead. All others are considered alive. Scale bars are 40 μm. **(B)** Quantification of the results shown in (A). Measurements are in biological triplicates with bars representing mean ± SD. Statistical analysis was performed using two-way ANOVA; ns: not significant; *$p < 0.05$, **$p < 0.005$. Pairwise comparisons across the subgroups did not show any significant differences. The underlying data can be found in Mendeley (see data statement for details).

finding that correlates well with earlier studies, which showed that while polyP chain lengths range widely across mammalian tissues, very long chain (>800 Pi units) polyPs appear to be the most prominent in mammalian brains [17]. Moreover, in contrast to other polyanionic amyloid modifiers such as glucosaminoglycans (i.e., heparin), which are primarily found in the extracellular matrix [40], brain polyP is present both inside and outside of cells as it is released by neurons and taken up by astrocytes [41]. Based on all these findings, we reasoned that polyP might be one of the few known physiologically relevant modifiers of disease-associated amyloid formation in the brain. Earlier findings that showed that polyP levels in the mammalian brain decrease with age [42] formed the foundation for our current working model that the age-associated decline in polyP levels might contribute to the age-specific onset and progression of these devastating diseases. It was therefore highly intriguing to note that recently solved cryo-EM structures of patient-derived amyloid fibrils (i.e., α-Syn, Aβ, and Tau) contained a "mystery density," which fit all of polyP's criteria; a continuous density along the fibril axis, highly negatively charged and in its dimension consistent with phosphate molecules [6,7,24]. So far, we have been unsuccessful in solving the cryo-EM structure of any polyP-associated amyloid formed in vitro due to their altered straightened morphology. Moreover, we are unable to directly visualize amyloid-associated polyP because it is not recognized by our polyP-binding probe, which interacts with the termini of polyP chains. We also considered it impossible to quantify any potential polyP associated with patient-derived fibrils as this method requires acid hydrolysis followed by inorganic phosphate determination which

requires extremely pure starting material. To address this important question, we hence decided to use an unbiased in silico approach to determine the potential polyP interface and use biochemical assays to validate the results. Based on these experiments, we now propose that polyP is the most plausible fit for the negatively charged density present in patient-derived fibrils.

In silico docking and MD simulation studies fit polyP-14 into the lysine-rich pocket of smaller structures of MSA-derived α-Syn fibrils and in vitro formed α-Syn protofilament interfaces. Moreover, docking simulations also sampled this position in the trimeric structure of the single protofilament structure with the shorter polyP-5 chain while no interaction at that site was predicted for the monomeric structures. These results are in excellent agreement with previous in vitro studies, which showed that polyP does not interact with monomeric α-Syn and hence does not affect the initial lag phase of fibril formation, yet significantly accelerates the process once smaller oligomers have formed [19]. Mutational studies allowed us to validate K43 and K45 as the major polyP interaction sites in α-Syn as replacing them with alanine residues was sufficient to abolish polyP binding, and, as a direct consequence, prevented polyP to (i) accelerate fibril formation; (ii) stabilize α-Syn fibrils; (iii) alter fibril morphology; and (iv) mitigate α-Syn cytotoxicity. Intriguingly, the substitution of K43 and K45 with uncharged residues mimicked some of the previously observed polyP effects, including an increase in the rate of fibril formation and stability. We now reason that the substitution of these clusters of positive charges lowers the energetic barrier of self-association as they experience less local electrostatic repulsion. PolyP binding to these oligomers serves a similar role by charge-neutralizing the core cluster of positive residues, thus accelerating fibril formation and increasing the cross-beta structure within each protofilament. In contrast, neither fibril morphology nor toxicity of the sonicated fibrils was substantially altered by the lack of these 2 residues, suggesting that polyP binding to the fibrils is necessary to promote these fibril properties.

It is noteworthy to point out that the 2 critical lysine residues K43 and K45 as well as H50, whose side-chains constitute the positive cluster associated with the mystery density binding in α-Syn, are part of a local hotspot known to harbor several mutations that elicit early onset familial Parkinson's disease. In fact, 3 disease-associated mutations, i.e., E46K, G51D, and A53T are part of this loop region [43]. Given that most of these mutations alter the local charge environment of this region, it is now tempting to speculate that they affect the electrostatic interactions with polyP, hence precluding the polyanion from modulating and stabilizing fibril formation, structure or both. Future studies will show whether polyP binding or lack thereof, plays a role in the early onset of this disease.

## Materials and methods

### Materials

Defined chain lengths of polyP-14, polyP-130, and polyP-300 were a kind gift from Dr. Toshikazu Shiba (Regenetiss, Japan). Unless noted otherwise, chemicals were purchased from Sigma-Aldrich.

### Molecular docking simulations with Autodock

The binding interaction between polyP-5, polyP-14, and α-Syn was studied using the open-source programs AutoDock 4.2 and Vina 1.1.2 [44]. To search for polyP binding sites, a linear 5-mer or 14-mer polyP structure was built using ChemDraw Professional 16.0 and energy was minimized with the Optimize Geometry function in Avogadro 1.2.0. The monomer and fibril structures of α-Syn were obtained from the Protein Data Bank (Table 1). Next, the polyP and α-Syn structures were parsed to the Python Molecular Viewer and Autodock Tools to generate the PDBQT files that add polar hydrogens, calculate partial atomic charges, and define atom

types. A grid box was manually adjusted around the α-Syn structure using the Molecular Graphics Laboratory (MGL) tool graphical user interface such that the protein is entirely embedded within the grid box for a blind docking simulation. The exhaustiveness was set to 8 and the remaining docking parameters were set to default.

## Molecular dynamics simulations using Desmond

To investigate the dynamic stability of the polyP—α-Syn complex, an all-atoms MD simulation of 6XYO, 6H6B, 6H6B-polyP, and 6XYO-polyP complex was performed, using the Desmond Molecular Dynamics System (Schrödinger, New York, NY, 2022.1). Using the system builder in Schrödinger suite, the complex was prepared using the TIP3P water model in an orthorhombic box with a dimension of 10 Å × 10 Å × 10 Å with periodic boundary conditions on the x, y, and z-axes. An ionic strength of 0.15 M NaCl was used to neutralize the complex system. The OPLS4 force field [35], the particle mesh Ewald (PME) was used to minimize the energy of the complex. A 9 Å cut-off was applied for Lennard-Jones and short-range coulombic interactions, and the PME method was used to simulate long-range electrostatic interactions [45]. Each system was subjected to equilibration in an equilibrium state at 1,000 steps with 100 ps time steps. The PME summation was used to calculate long-range electrostatic interactions. The system's production step was run for 100 ns for each system (6XYO, 6H6B, 6H6B-polyP, and 6XYO-polyP complex) utilizing the Nose–Hoover technique with an NPT ensemble with a time step of 100 ps, temperature of 310 K, and pressure of 1.0 atm. The harvested trajectories were analyzed using a simulation interaction diagram and simulation event analysis tools implemented in the Desmond MD package [46].

## All-atoms MD simulation using Gromacs

All-atom MD simulations of polyP interactions were performed with monomeric α-Syn (8A9L), type-I α-Syn filament (6XYO), and α-Syn fibrils (6H6B) using Gromacs-v2021.4 [32]. To reveal the interaction of polyP-14 with α-Syn using MD simulation, the polyP chain was initially placed >1 nm away from the protein. The protonation states of each protein were set to pH 7.0 using the protein-preparation wizard in Maestro. The Charmm36 force [33] field was used for proteins, while polyP was parameterized using Charmm forcefield in SwissParam [47,48]. Each system was solvated in a periodic cubic water box of 10 nm using the TIP3P water model. Upon solvation, systems were subjected to electro-neutralization with the addition of $Cl^-$ ions and $Na^+$ ions, using an ionic strength of 0.15 M. Next, the electro-neutralized systems were subjected to energy minimization to remove steric clashes or bad contacts using the steepest descent algorithm for 10,000 steps, until the maximum force was abridged to less than 10 kJmol$^{-1}$. Following energy minimization, the systems were equilibrated using NVT (i.e., a fixed number of simulated particles (N), volume (V), and temperature (T) throughout the simulation) and NPT (i.e., a fixed number of simulated particles (N), pressure (P), and temperature (T) throughout the simulation) ensembles. The NVT ensemble for 2 ns at 310 K was done using the velocity-rescaling method, whereas NPT equilibration was done at 1 bar for 5 ns using the Berendsen barostat. A cut-off value for the neighbor list at 1 nm was used for non-bonded interactions and a cut-off of 1.0 nm was used to handle van der Waals interactions using the potential-shift-Verlet approach. Electrostatic interactions were defined using the PME method. The hydrogen bonds were constrained using a LINCS algorithm with a time step of 2 fs. The production MD was conducted for 50 ns for α-Syn fibrils and 100 ns for α-Syn monomer interaction with polyP-14, using the velocity-rescaling method to control temperature and a Parinello–Rahman barostat to maintain pressure. The coordinates are saved every 10 ps and used for post dynamics analysis.

## Post-MD analysis

To explore the effect of polyP-14 binding on α-Syn, the MD trajectories were deconvoluted to extract dynamics stability statistics including the Cα root mean square deviations and fluctuations, polyP contact maps, and intermolecular bonding of each MD system.

## Cloning and purification of α-Syn mutants

The α-Syn WT containing plasmid pT7-7 was mutagenized using the QuickChange II site-directed mutagenesis protocol (Agilent). The primers used for the mutagenesis can be found in S1 Table. Successful mutagenesis was confirmed via Sanger Sequencing (GENEWIZ). Expression and purification of WT α-Syn and the mutants were performed as previously described [18]. Briefly, BL21(DE3) *Escherichia coli* strains expressing either WT or mutant α-Syn from a pT7-7 vector plasmid were grown in Luria broth supplemented with 200 μg/ml ampicillin until $OD_{600}$ of 0.8 to 1.0 was reached. Protein expression was induced with 0.8 mM IPTG for 4 h, after which bacteria were harvested at 4,500×g for 20 min and 4˚C. The pellet was resuspended in 50 ml lysis buffer (10 mM Tris-HCl (pH 8.0), 1 mM EDTA, Roche Complete protease inhibitor cocktail) and the lysate was boiled for 20 min. Aggregated proteins were removed by centrifugation at 13,500×g for 30 min at 4˚C. The supernatant was removed and combined with 136 μl/ml of a 10% w/v solution of streptomycin sulfate and 228 μl/ml glacial acetic acid. After a further centrifugation step at 13,500×g for 30 min at 4˚C, the supernatant was removed and mixed in a 1:1 ratio with saturated ammonium sulfate and incubated with occasional stirring at 4˚C for 1 h. Next, the mixture was centrifuged at 13,500×g for 30 min at 4˚C and the pellet was resuspended in 10 mM Tris-HCl (pH 7.5). Using concentrated NaOH, the pH of the suspension was adjusted to pH 7.5 and the protein was dialyzed against 10 mM Tris-HCl (pH 7.5), 50 mM NaCl overnight. Next, the protein was filtered and loaded onto 2 connected 5 ml HiTrap Q HP columns (GE Healthcare, 17-1154-01) equilibrated with 10 mM Tris-HCl (pH 7.5), 50 mM NaCl. Washing steps were performed with 10 mM Tris-HCl (pH 7.5), 50 mM NaCl and the protein was eluted with a linear gradient from 50 to 500 mM NaCl. The protein-containing fractions were combined and dialyzed against 40 mM KPi, 50 mM KCl (pH 7.5). Oligomeric α-Syn species were removed by filtering the protein through a 50-kD cutoff column (Amicon, Millipore). Aliquots of the protein were prepared, lyophilized, and stored at −80˚C. No adjustments were made for any of the mutants.

## Thioflavin T fluorescence

A total of 100 μM freshly purified α-Syn monomers were incubated with 10 μM thioflavin T (ThT) in 40 mM KPi, 50 mM KCl (pH 7.5) at 37˚C and two 2-mm borosilicate glass beads in the absence or presence of 500 μM (given in Pi units) polyP-14, polyP-130, or polyP-300. For ThT-measurements, samples were pipetted into black 96-well polystyrene microplates with clear bottoms (Greiners). ThT fluorescence was detected in 10 min intervals for 72 h using a Synergy HTX MultiMode Microplate Reader (Biotec) with an excitation of 440 nm, emission of 485 nm, and a gain of 35.

## Dissociation of α-Syn fibrils by acid hydrolysis and polyP determination using molybdate assay

Fibril formation was achieved as described above using 100 μM of WT or mutant α-Syn in the absence or presence of increasing concentrations of polyP-130 in a volume of 100 μl. Mature fibril formation was verified by ThT measurements (>48 h). The fibrils were pelleted at 20,000×g for 20 min and washed with 50 mM Tris-HCl (pH 7.5), 150 mM NaCl to remove

excess polyP, salts, and small oligomers. The pellets were subjected to washes with a high salt buffer (50 mM Tris-HCl (pH 7.5), 500 mM NaCl) to reduce nonspecific interactions and a low salt buffer (50 mM Tris-HCl (pH 7.5), 150 mM NaCl) to remove excess salts before resuspending in 50 μl 1 M HCl to release all remaining polyP as free phosphate (Pi). The samples were incubated for 1 h at 95˚C and neutralized with equal volumes of 1 M NaOH. The polyP concentrations were determined using the calorimetric molybdate assay [38]. In brief, 75 μl of a freshly prepared working solution prepared by mixing 91.2 parts (v/v) of a detection base containing 600 mM sulfuric acid, 2.4 mM ammonium heptamolybdate, 600 μM antimony potassium tartarate, and 88 mM ascorbic acid and 8.8 parts (v/v) of 1 M ascorbic acid was added to 25 μl of sample. Samples were incubated at room temperature for 5 min before measuring absorbance at 882 nm. The concentration of Pi was calculated according to a standard curve using potassium phosphate (KPi).

### SDS stability measurements

A total of 100 μM WT or mutant α-Syn fibrils formed in the absence or presence of 500 μM polyP-130 were resuspended in 50 mM Tris-HCl, 150 mM NaCl (pH 7.5), 20 μM ThT. Next, samples were supplemented with increasing concentrations of sodium dodecyl sulphate (SDS) and incubated at 37˚C in a double orbital shaker at 600 rpm for 5 min. ThT fluorescence was measured as previously stated as a measure of fibril stability normalized against initial ThT signals for samples with 0% SDS [39]. Alternatively, the fibrils were supplemented with SDS to a final concentration of 0.2% w/v and incubated at 37˚C in a double orbital shaker at 600 rpm. Samples were taken at the indicated times (S7 Fig) and pelleted at 4˚C by centrifugation (16,000×g, 10 min). The supernatant fraction was run on a 12% SDS PAGE and stained with Coomassie brilliant blue. The amount of protein in the supernatant was quantified by densitometry and normalized against the total input protein fraction.

### Negative stain of fibrils and transmission electron microscopy (TEM) analysis

For TEM experiments, 100 μM WT or mutant α-Syn in the absence or presence of 500 μM polyP-130 was used to form fibrils following the same protocol as above for ThT fluorescence experiments (without adding ThT). Samples were negatively stained with 0.75% uranyl formate (pH 5.5 to 6.0) on thin amorphous carbon layered 400-mesh copper grids (Pelco) as has been reported previously [13]. In brief, 5 μl of the sample was applied onto the grid and left for 3 min before removing excess liquid with Whatman paper. The grid was washed twice with 5 μl ddH2O followed by 3 applications of 5 μl uranyl formate. Grids were imaged at room temperature using a Fei Tecnai 12 microscope operating at 120 kV. Images were acquired on a US 4,000 CCD camera at 66,873× resulting in a sampling of 2.21 A/pixel, and 10 micrographs were taken for each sample in a given experiment and 3 individual experiments were carried out, resulting in a total of 30 micrographs per sample. In total, at least 100 individual α-Syn filaments were selected across micrographs. The filament widths were determined using the micrograph dimensions as a reference and pixel widths were converted into angstroms using the program ImageJ. While measuring filament widths the researcher was blind to which sample the micrograph belonged to.

### Cell culture toxicity assays

SH-SY5Y neuroblastoma cells were differentiated using previously established protocols [49] and were seeded onto a cell culture-treated black 96-well plate with a density of 15,000 cells per well in 100 μl differentiation media (Neurobasal, 1xB27, 1xPen/Strep, 50 ng/ml BDNF,

and 10 µM retinoic acid). To freshly differentiated cells, 4 µM sonicated fibrils of WT or mutant α-Syn in the presence of 0 or 25 or 50 µM polyP-130 were supplemented and incubated for 24 h before assaying for viability.

## Statistical analysis

One-way ANOVA with post hoc analysis was used to compare the various groups. $P$-values $<0.05$ were considered significant. All data in the bar charts are displayed as mean ± SD. Replicate numbers ($n$) are listed in each figure legend. Prism 7.04 (GraphPad) was used to perform statistical analysis.

## Supporting information

**S1 Fig. Cryo-EM structures of α-Syn polymorphs mapped with electron densities. (A)** Density map of patient-derived α-Syn fibrils of the MSA polymorph (PDB: 6XYO), consisting of 2 protofilaments as indicated by the blue and green backbone structures. **(B)** Density map of patient-derived α-Syn fibrils of the Lewy-fold polymorph (PDB: 8A9L), which consist of a single protofilament. **(C)** Density map of in vitro-derived α-Syn fibrils (PDB: 6H6B), which consist of 2 protofilaments as indicated by the blue and green backbone structures. A central nonproteinaceous "mystery density" is found in both patient-derived fibrils as indicated by the arrows in 6XYO and 8A9L but is absent in the in vitro-derived α-Syn structure 6H6B. The mystery density is surrounded by residues K43, K45, and His50 in 6XYO and K32, K34, K43, and K45 in 8A9L. The cartoon structures were generated using ChimeraX program. The underlying data can be found in Mendeley (see data statement for details).
(DOCX)

**S2 Fig. Structures of α-Syn monomers and selected fibril polymorphs and their molecular interaction with polyP. (A)** PDB structures of an 8A9L monomer, a micelle-bound 1XQ8 monomer, and the α-Syn fibrils 8A9L, 6XYO, and 6H6B. **(B)** Docked structure of polyP-14 (ball and stick) to α-Syn monomers or fibrils obtained using AutoDock Vina blind docking simulation. PolyP-14 binding residues are labeled and hydrogen bonds are indicated as dashed lines. The cartoon structures were generated using Discovery Studio Visualizer and ChimeraX programs. The underlying data can be found in Mendeley (see data statement for details).
(DOCX)

**S3 Fig. Docking of polyP into the cryo-EM structures of MSA type-I filaments of α-Syn and Lewy-fold single protofilament. (A)** A blind docking simulation of polyP-14 (in yellow) matches the unknown electron density in 6XYO. A zoom surface structure prepared using Discovery Studio Visualizer shows the polyP-14 binding pocket fits into the missing non-proteaceous high charge density core. **(B)** The docked structure of polyP-5 (red) to the Lewy-fold single protofilament structure 8A9L shows its occupancy within the mystery density. The cartoon structure was generated using the ChimeraX program. The underlying data can be found in Mendeley (see data statement for details).
(DOCX)

**S4 Fig. MD analysis assessing the structural and dynamic properties of polyP-14 and α-Syn fibrils. (A)** Interaction plots between polyP-14 with α-Syn obtained during 100 ns MD simulations using Desmond. The stacked bar plots are normalized over the course of the trajectory where a value of 0.5 suggests that a specific interaction is maintained 50% of the simulation time. A value over 1.0 indicates these α-Syn residues make multiple contacts of the same subtype with the polyP. **(B)** A timeline representation of α-Syn interaction with polyP-14 over the 100 ns MD simulation. The total contacts between α-Syn and polyP-14 are shown in the

top panel. Chain names for individual residues are indicated with a colon. **(C)** Stability of αSyn-polyP-14 complex measured as a function of protein Cα, and polyP all-atoms root mean square deviation with respect to 100 ns MD simulation using Desmond. **(D)** Dynamics and solvent accessible properties of polyP-14 complexed with 6XYO obtained from 100 ns MD simulation. The underlying data can be found in Mendeley (see data statement for details). (DOCX)

**S5 Fig. Interaction of polyP-14 and in vitro-derived α-Syn polymorph 6H6B.** MD snapshots showing the polyP-14 interaction with the cryo-EM structure of in vitro-derived α-Syn fibrils (PDB ID: 6H6B) before and after 50 ns MD simulation. Cartoon structures shown at the bottom presents the polyP-14 interacting residues in α-Syn 6H6B structure obtained from Auto-Dock molecular docking simulation (left) with grid center surrounding the binding pocket involving residues K43 and K45, and at 50 ns MD simulation using Gromacs (right). Chain names for individual residues are indicated with a colon. The underlying data can be found in Mendeley (see data statement for details). (DOCX)

**S6 Fig. Root mean square fluctuation (RMSF) plots for individual αSyn molecules in 6XYO and 6H6B, complexed with and without polyP-14 as indicated obtained from 100 ns MD simulation using Desmond.** Arrows point to the α-Syn chain numbers complexed with polyP-14 that exhibit higher fluctuations in the 50–60 residue regions (highlighted in blue) compared to the fibers alone. Residues shaded in gray indicate regions of high RMSF in α-Syn fibers both with and without polyP-14. The underlying data can be found in Mendeley (see data statement for details). (DOCX)

**S7 Fig. Effects of K43, K45 substitution on fibril stability.** Fibrils were prepared in the absence or presence of 500 µM polyP-130 as in Fig 3C. Fibrils were supplemented with SDS to a final concentration of 0.2% (w/v) and pelleted at the indicated time points. The supernatant was loaded onto an SDS PAGE. The total protein was set to 100%; $n = 3$; mean ± SEM is shown. The underlying data can be found in Mendeley (see data statement for details). (DOCX)

**S1 Table. Primers used for site directed mutagenesis of αSyn.** (DOCX)

## Acknowledgments

We thank the members of the Jakob and Bardwell labs for many helpful discussions. We thank Ke Wan for purifying the proteins used in this study and Dr. Toshikazu Shiba (Regenetiss, Japan) for providing us with purified polyP. We thank the University of Michigan electron microscopy facility for their training and access to the transmission electron microscope.

## Author Contributions

**Conceptualization:** Philipp Huettemann, Justine Lempart, Ursula Jakob.

**Formal analysis:** Philipp Huettemann, Pavithra Mahadevan, Budheswar Dehury, Daniel R. Southworth, Bikash R. Sahoo.

**Funding acquisition:** Ursula Jakob.

**Investigation:** Philipp Huettemann, Pavithra Mahadevan, Justine Lempart, Eric Tse, Budheswar Dehury, Brian F. P. Edwards, Bikash R. Sahoo.

**Methodology:** Pavithra Mahadevan.

**Project administration:** Bikash R. Sahoo, Ursula Jakob.

**Resources:** Brian F. P. Edwards, Bikash R. Sahoo.

**Supervision:** Daniel R. Southworth, Bikash R. Sahoo, Ursula Jakob.

**Validation:** Pavithra Mahadevan.

**Writing – original draft:** Philipp Huettemann.

**Writing – review & editing:** Ursula Jakob.

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
