## [Editor Report · Decision Letter 0]

30 Apr 2024

Dear Ursula, 

Thank you very much for submitting your manuscript entitled "Amyloid Accelerator Polyphosphate Implicated as the Mystery Density in α-Synuclein Fibrils" for consideration as a Short Report by PLOS Biology. Please accept my sincere apologies for the delay in getting back to you as we consulted with an academic editor about your submission.

Your manuscript has now been evaluated by the PLOS Biology editorial staff, as well as by an academic editor with relevant expertise, and I am writing to let you know that we would like to send your submission out for external peer review.

Once your full submission is complete, your paper will undergo a series of checks in preparation for peer review. After your manuscript has passed the checks it will be sent out for review. To provide the metadata for your submission, please Login to Editorial Manager (https://www.editorialmanager.com/pbiology) within two working days, i.e. by May 02 2024 11:59PM.

Best wishes,

Richard

Richard Hodge, PhD

rhodge@plos.org

PLOS

---

## [Decision Letter · Decision Letter 1]

13 Jun 2024

Dear Dr Jakob,

Thank you for your patience while your manuscript "Amyloid Accelerator Polyphosphate Implicated as the Mystery Density in α-Synuclein Fibrils" was peer-reviewed at PLOS Biology. It has now been evaluated by the PLOS Biology editors, an Academic Editor with relevant expertise, and by several independent reviewers. 

In light of the reviews, which you will find at the end of this email, we would like to invite you to revise the work to thoroughly address the reviewers' reports.

As you will see below, the reviewers comment that the study makes an important contribution to the field and highlight that it is generally a well done. However both reviewers have raised a number of important suggestions to improve the presentation of the work and to strengthen the study further. We think that these should be carefully addressed before publication. While many of the reviewer comments require only textual changes, I would like to emphasize that we think it will be important to provide the appropriate controls and other data requested by reviewer 2, to further support the conclusions of the work.

Given the extent of revision needed, we cannot make a decision about publication until we have seen the revised manuscript and your response to the reviewers' comments. Your revised manuscript is likely to be sent for further evaluation by all or a subset of the reviewers.

**IMPORTANT - SUBMITTING YOUR REVISION**

*Re-submission Checklist*

*Published Peer Review*

*PLOS Data Policy*

*Blot and Gel Data Policy*

Sincerely,

Suzanne

Suzanne De Bruijn, PhD, 

Associate Editor

PLOS Biology

sbruijn@plos.org

REVIEWS:

Reviewer #1: Jacobs et al. present a very convincing study of the binding of polyphosphate polymers to alpha-synuclein fibrils both in silico and in vitro with effects in cellula. They identify potential binding sites on different a-syn fibrils, show that they are compatible with hitherto unassigned "mystery densities" in cryoEM maps and validate this by rational mutagenesis. This sheds striking new light on an important physiologically relevant phenomenon and provides a strong case for the basis of the known binding to fibrils. I only have minor points to suggest:

Minor points:

1. Overall provide more information on polyP biology: what is the typical (if any known) molecular weight distribution, how do they form and what is known about their role. Would we expect different polymer lengths to have different effects or is it (as e.g. heparin) largely a question of having a minimum length to promote binding etc. Overall polyanions have remarkable effects in biology - are there any connections to e.g. heparin or is that an unrelated phenomenon as heparin like many other glucosaminoglycans is extracellular?

2. Fig. S1: It would be helpful to the reader if the two alpha-syn monomers in the structures in panels A and B could be shown in separate colors. Currently it is a bit difficult to make out where the interface between the two protofilaments is. Furthermore, the mystery density could be shown in a different color for ease of visualization.

3. Fig. S2: why only use 3 a-syn monomers for 8A9L but 5 for 6XYO and 6H6B?

4. Fig. 1AB and Fig. S2B: Could the two polyP chains be shown in different colors for ease of visualization? Use same colors in panels A and B and other relevant figures.

5. Fig. S2B: No mystery density is shown so it is difficult to evaluate that polyP binds to 6H6B's mystery density but not that of 8A9L.

6. Figure legend figure 1: H=>G.

7. "hydrophobic interactions are shown to have minimal contribution in polyP binding to α-Syn fibrils with an average of ~48 total contacts (Fig. S4B)." As far as I can see from panel A, there are plenty of green (hydrophobic) itneractions and I simply can't read any hydrophobic contacts into panel B.

8. "three α-Syn molecules in 6XYO (Chains G, C, E) showed stronger root means square fluctuation in the central region (residues 50-60) in the presence of polyP when compared to 6XYO structure alone." This is definitely intriguing - but could it be an edge/end effect? (i.e. monomer units are less constrained at the ends of fibrils - after all the fibril is not infinitely long but only has 5 units for practical computational reasons).

9. Fig. 6B: x-axis lacks a proper "µM".

Reviewer #2: In this study, the authors modeled the binding of polyphosphate (polyP) to different alpha-synuclein (aSyn) polymorphs via docking studies and MD simulations. The results suggested that polyP can bind in a lysine-rich pocket previously shown to coordinate an uncharacterized electron-rich density in the structures of aSyn fibrils isolated from the brains of patients. Additional experiments revealed that two lysine residues within the lysine-rich pocket (K43 and K45) are necessary for effects of polyP on aSyn fibrillization, as well as on fibril stability, morphology, and cytotoxicity. The authors conclude that polyP modulates these properties of aSyn by alleviating electrostatic repulsion between lysine residues within the lysine-rich pocket.

This paper is significant in terms of its focus on the structural properties of aSyn amyloid-fibrils, which play a key role in the spread of pathology in the brains of individuals with PD and other synucleinopathy disorders. The demonstration that polyP fits within the mystery electron density observed in cryoEM structures of patient-derived fibrils, and that residues K43 and K45 play a central role in polyP-fibril interactions, represents an important breakthrough that will be of great interest to the PD field.

Strengths of the paper include its comprehensive scope (covering docking and MD simulations, biochemical analyses, and cell culture studies), the rigorous nature of the experimental design, and the high quality of the data. The statistical analyses are appropriate, the methods' section is written in sufficient detail to enable others to replicate the findings, and the supplementary information provides additional useful insights. On the other hand, parts of the manuscript lack details necessary for the reader's understanding, and several errors in the text detract from the overall quality of the paper. The paper would also be improved by adding key points to the discussion and by including additional data and controls. These issues should be addressed in a revised version of the manuscript.

p. 4, 'Transmission electron microscopy (TEM) and cryo-EM analysis revealed that polyP-associated alpha-Syn fibrils consist of a single protofilament with a reduced helical twist'; it should be specified that these were recombinant aSyn fibrils, and reference PMID 31533964 should be cited.

p. 5, 'These approaches have been successfully used to characterize the interaction of alpha-Syn with small molecules, membranes, metal ions, and other proteins (27, 28)'; this section would be improved by including citations to earlier studies involving MD simulations of the binding of small molecules to aSyn fibrils.

p. 5, 'as well a'; change to 'as well as a'.

p. 6, 'the simulations yield in a stronger binding affinity'; delete 'in'.

p. 6, 'coincides with the strongly electron negative density in 6XYO'; reword (e.g., change to 'strong electron density').

p. 6, 'each Pi-unit is in contact with 3-4 alpha-Syn molecules from both protofilaments while the free phosphate units at the ends of the polyP-14 structure interact with the first alpha-Syn molecule in each protofilament'; this is an interesting point that isn't evident in the images shown in Fig. 1. It would be helpful to include a panel that better illustrates this point.

p. 7, 'L-shape grove region'; replace with 'L-shaped groove region'.

p. 7, 'we found no interactions between polyP and alpha-Syn monomer 8A9L (Fig. 1D)'; for clarity, it would be helpful to indicate whether this part of the analysis involved polyP-5 or -14.

p. 8, 'K45 presented in the D/F alpha-Syn subunits of the one protofilament'; change 'presented' to 'present', and delete 'the' from 'the one protofilament'.

p. 8, 'Interaction network analysis obtained from 100 ns MD simulation of the alpha-Syn and polyP-14 complex in Desmond revealed that'; it should be specified which fibril structure was examined in this phase of the study.

p. 8, 'MD simulations using the in vitro generated polymorph (6H6B) yielded similar results and, as before, depicted polyP to interact with K43 and K45 residues distributed in one of the protofilaments with additional interactions that involve residues L38, Y38, and S42'; it is unclear why these results differed from those obtained from the docking analysis of 6H6B (i.e., why was polyP binding to the L-shaped groove region not seen here?) Also, the interaction between polyp and K45 is not illustrated in Fig. S5. The figure should be updated accordingly, or the reference to K45 should be deleted from the text. Lastly, 'Y38' should be changed to 'Y39'.

p. 9, 'Based on these results, we now speculate that this increase in root means square fluctuation may indicate an onset of disorder that could destabilize and alter the molecular packing or both within the β-stacked fibril morphology and might explain why polyp binding to in vitro formed double-twisted fibrils leads to their dissociation into individual protofilaments (19)'; in light of the discussion that follows, this statement would be more convincing if it were shown that the increase in root mean square fluctuations also occurs with the fibril structure of recombinant aSyn, 6H6B'.

p. 9, 'a ratio of polyP (in Pi-units) to alpha-Syn monomer of 5:1'; the use of a 5:1 ratio should be justified (e.g., in terms of physiologically relevant aSyn and polyP concentrations). In addition, it would be helpful if the authors included data for mixtures with a lower polyP-aSyn ratio, if these data are available.

p. 11, 'had no significant affect the fibril morphology of the mutant variant'; change to 'had no significant effect on the fibril morphology …'

p. 12, 'After 5 min of incubation, we monitored the changes in ThT fluorescence as a measure of fibril destabilization and dissociation'; the data are difficult to interpret in light of potential quenching effects of SDS on ThT fluorescence. This question could be addressed by performing fluorescence measurements immediately after adding the SDS, or by using a different method to monitor aSyn fibril levels such as ultracentrifugation coupled with Western blot analysis.

p. 12, 'unaffected by the presence polyP'; change to 'presence of'.

p. 14, 'PolyP binding to these oligomers serves a similar role by charge-neutralizing the core cluster of positive residues, thus accelerating fibril formation and increasing fibril stability'; this statement might seem confusing in light of the earlier statement that 'an onset of disorder that could destabilize and alter the molecular packing or both within the beta-stacked fibril morphology' (p. 9). This part of the discussion would be improved by clarifying that polyP destabilizes interactions between protofilaments but stabilizes the cross-beta structure within each protofilament.

p. 15, 'four disease-associated mutations, i.e., E46K, H50Q, G51D, and A53T'; genetic data suggest that H50Q is not a pathogenic variant of aSyn (e.g., see PMID 29398121).

p. 17, 'fixed number of simulated particles (P), volume (V) and temperature (T) throughout the simulation) and NPT (i.e., a fixed number of simulated particles (P), pressure (P) and temperature (T) throughout the simulation) ensembles'; presumably the 'number of simulated particles' should have the abbreviation 'N'?

p. 21, 'In vivo toxicity assays'; suggest replacing 'in vivo' with 'cell culture'.

Fig. 1G, legend: the letter should be changed from 'H' to 'G', and the identity of the aSyn structure should be indicated.

Fig. 3C, legend: 'Fibrils of WT alpha-Syn (black/grey) or α-Syn [K43A,K45A mutant] (purple)'; replace 'purple' with 'light and dark purple'. 

Fig. 4B: the data would be more convincing if additional pairwise comparisons were presented (e.g., WT aSyn at 0 versus 25 micromolar or 0 versus 50 micromolar polyP-130), and for this, a two-way ANOVA could be useful.

Fig. S4 legend: the aSyn structure should be specified, and it should be indicated that the results were obtained using the Desmond method. Also, in panel A, it is uncertain how the data were normalized. It is stated that a value of 0.5 corresponds to a given interaction being maintained 50% of the simulation time, but then it's unclear why certain segments representing different interaction types have lengths >1. This point should be addressed.

---

## [Decision Letter · Decision Letter 2]

14 Aug 2024

Dear Ursula,

Thank you for your patience while we considered your revised manuscript "Amyloid Accelerator Polyphosphate Implicated as the Mystery Density in α-Synuclein Fibrils" for publication as a Short Report at PLOS Biology. This revised version of your manuscript has been evaluated by the PLOS Biology editors, the Academic Editor and the original reviewers.

Based on the reviews, I am pleased to say that we are likely to accept this manuscript for publication, provided you satisfactorily address the remaining comments raised by Reviewer #2. Please also make sure to address the following data and other policy-related requests that I have provided below (A-E):

(A)We would like to suggest the following minor modification to the title:

“Amyloid accelerator polyphosphate is the mystery density observed in structures of α-Synuclein Fibrils and enhances fiber stability"

(B) You may be aware of the PLOS Data Policy, which requires that all data be made available without restriction: http://journals.plos.org/plosbiology/s/data-availability. For more information, please also see this editorial: http://dx.doi.org/10.1371/journal.pbio.1001797

-Supplementary files (e.g., excel). Please ensure that all data files are uploaded as 'Supporting Information' and are invariably referred to (in the manuscript, figure legends, and the Description field when uploading your files) using the following format verbatim: S1 Data, S2 Data, etc. Multiple panels of a single or even several figures can be included as multiple sheets in one excel file that is saved using exactly the following convention: S1_Data.xlsx (using an underscore).

-Deposition in a publicly available repository. Please also provide the accession code or a reviewer link so that we may view your data before publication. 

Figure 2A-B, 3A-C, 4B, S4A-D, S6

(C) Please also ensure that each of the relevant figure legends in your manuscript include information on *WHERE THE UNDERLYING DATA CAN BE FOUND*, and ensure your supplemental data file/s has a legend.

(D) Please ensure that your Data Statement in the submission system accurately describes where your data can be found and is in final format, as it will be published as written there. 

(E) Per journal policy, if you have generated any custom code during the course of this investigation, please make it available without restrictions. Please ensure that the code is sufficiently well documented and reusable, and that your Data Statement in the Editorial Manager submission system accurately describes where your code can be found. 

Please note that we cannot accept sole deposition of code in GitHub, as this could be changed after publication. However, you can archive this version of your publicly available GitHub code to Zenodo. Once you do this, it will generate a DOI number, which you will need to provide in the Data Accessibility Statement (you are welcome to also provide the GitHub access information). See the process for doing this here: https://docs.github.com/en/repositories/archiving-a-github-repository/referencing-and-citing-conten

We expect to receive your revised manuscript within one month. 

*Published Peer Review History*

*Press*

Best wishes,

Richard

Richard Hodge, PhD

rhodge@plos.org

Reviewer remarks:

Reviewer #1: The authors have addressed my concerns very well.

Reviewer #2: The authors have responded well to the initial comments, and as a result the paper is much improved.

There are a few remaining issues that should be addressed in a revised version of the manuscript:

p. 7, 'L-shaped grove region'; replace 'grove' with 'groove'.

Fig. 1B, legend, 'The two free phosphate units interacting with the first two alpha-Syn molecules facing opposite in each protofilament (chains G and H) are shown in pink lines'; replace with 'Shown are interactions (represented by pink lines) representing one free phosphate unit interacting with the first two alpha-Syn molecules facing opposite in each protofilament (chains G and H)'.

Fig. 1G legend: the identity of the aSyn structure analyzed should be specified. For Fig. S4, 6XYO should be specified earlier in the legend.

Fig. 4B: the two-way ANOVA should be used to assess whether additional pairwise comparisons across subgroups defined by different polyp-130 concentrations (e.g., WT aSyn at 0 versus 25 micromolar or 0 versus 50 micromolar polyP-130) show significant differences.

Response to the question about SDS effects on ThT fluorescence:

p. 12, 'After 5 min of incubation, we monitored the changes in ThT fluorescence as a measure

of fibril destabilization and dissociation'; the data are difficult to interpret in light of potential

quenching effects of SDS on ThT fluorescence. This question could be addressed by

performing fluorescence measurements immediately after adding the SDS, or by using a

different method to monitor aSyn fibril levels such as ultracentrifugation coupled with Western

blot analysis.

While we appreciate this concern, we do not think that additional experiments would be

necessary as any potential quenching effects of SDS on ThT fluorescence would be the same

for all samples.

This statement is unconvincing because SDS could have have differential effects on ThT binding to various fibrillar polymorphs at concentrations of the detergent below those required to induce fibril dissociation. In addition, the extent of fibril dissociation in SDS solutions after only 5 minutes of incubation at 37 deg C is uncertain. Given the limitations of quantifying aSyn fibrils solely via ThT fluorescence measurements, the results would be strengthened if complemented by a different method to monitor aSyn fibril levels, such as ultracentrifugation coupled with Western blot analysis.

---

## [Editor Report · Decision Letter 3]

23 Sep 2024

Dear Ursula,

On behalf of my colleagues and the Academic Editor, Raquel Lieberman, I am pleased to say that we can accept your manuscript for publication, provided you address any remaining formatting and reporting issues. These will be detailed in an email you should receive within 2-3 business days from our colleagues in the journal operations team; no action is required from you until then. Please note that we will not be able to formally accept your manuscript and schedule it for publication until you have completed any requested changes.

PRESS

Best wishes, 

Richard Hodge, PhD

rhodge@plos.org

PLOS
